# *Alu*-dependent RNA editing of GLI1 promotes malignant regeneration in multiple myeloma

Elisa Lazzari[1], Phoebe K. Mondala[1], Nathaniel Delos Santos[1], Amber C. Miller[2], Gabriel Pineda[1,3], Qingfei Jiang[1], Heather Leu[1], Shawn A. Ali[1], Anusha-Preethi Ganesan[1], Christina N. Wu[1], Caitlin Costello[4], Mark Minden[5], Raffaella Chiaramonte[6], A. Keith Stewart[2], Leslie A. Crews[1] & Catriona H.M. Jamieson[1,4]

Despite novel therapies, relapse of multiple myeloma (MM) is virtually inevitable. Amplification of chromosome 1q, which harbors the inflammation-responsive RNA editase adenosine deaminase acting on RNA (ADAR)1 gene, occurs in 30–50% of MM patients and portends a poor prognosis. Since adenosine-to-inosine RNA editing has recently emerged as a driver of cancer progression, genomic amplification combined with inflammatory cytokine activation of ADAR1 could stimulate MM progression and therapeutic resistance. Here, we report that high ADAR1 RNA expression correlates with reduced patient survival rates in the MMRF CoMMpass data set. Expression of wild-type, but not mutant, ADAR1 enhances Alu-dependent editing and transcriptional activity of GLI1, a Hedgehog (Hh) pathway transcriptional activator and self-renewal agonist, and promotes immunomodulatory drug resistance in vitro. Finally, ADAR1 knockdown reduces regeneration of high-risk MM in serially transplantable patient-derived xenografts. These data demonstrate that ADAR1 promotes malignant regeneration of MM and if selectively inhibited may obviate progression and relapse.

---

[1] Division of Regenerative Medicine, Department of Medicine, Moores Cancer Center, University of California, San Diego, La Jolla, CA 92037, USA. [2] Department of Medicine, Mayo Clinic, Rochester, MN 55905, USA. [3] Department of Health Sciences, School of Health and Human Services at National University, San Diego, CA 92123, USA. [4] Department of Medicine, Moores Cancer Center at University of California, San Diego, La Jolla, CA 92093, USA. [5] Princess Margaret Hospital, University Health Network, Toronto, ON, Canada M5G 2M9. [6] Department of Health Sciences, University of Milan, Milan 20142, Italy. Correspondence and requests for materials should be addressed to L.A.C. (email: lcrews@ucsd.edu) or to C.H.M.J. (email: cjamieson@ucsd.edu)

Multiple myeloma (MM) is a plasma-cell neoplasm that represents the second most common blood cancer in the United States. High-risk forms of this disease have been associated with amplifications at chromosome 1q21, which occurs in over 30% of MM patients and is associated with poor clinical outcomes[1,2]. Notably, the inflammation-responsive RNA editase gene adenosine deaminase acting on RNA-1 (ADAR1) as well as interleukin-6 receptor (IL-6R) localize to this unique

**Fig. 1** ADAR1 is overexpressed in high-risk, advanced MM patients. **a** Representative diagram of *ADAR* genomic locus on chromosome 1q21. **b** ADAR1 relative expression (Fragments Per Kilobase of transcript per Million mapped reads, FPKM) in primary CD138[+] cells from patients harboring 1q amplifications, compared to no 1q (1q stage I $n = 29$, stage II $n = 39$, stage III $n = 32$; no 1q stage I $n = 146$, stage II $n = 150$, stage III $n = 133$). The error bars represent ± S.E.M. of the mean; **$p < 0.01$ by two-tailed, Mann–Whitney U test. **c** IL6R relative expression in primary CD138[+] cells from patients harboring 1q amplifications, compared to no 1q. The error bars represent ±S.E.M. of the mean; **$p < 0.01$ by two-tailed, Mann–Whitney U test. **d** ADAR1 isoform relative expression in 1q-amplified (1q) versus no 1q patients. The error bars represent ±S.E.M. of the mean; *$p < 0.05$, **$p < 0.01$ by two-tailed, Mann–Whitney U test. **e** Total ADAR1 mRNA levels assessed in total MNCs from primary samples ($n = 19$; see Table 1). Age-matched ($n = 3$, mean age = 60.6 ± 16.8 years old) BM collected from patients undergoing hip replacement therapy for reasons other than cancer were used as normal healthy controls. HPRT gene expression was used for normalization. Dots represent mean values for individual patients ± S.E.M (ctrl $n = 3$; smoldering MM $n = 4$; newly diagnosed MM $n = 4$; relapsed MM $n = 7$; PCL $n = 4$). **$p < 0.01$ compared to normal controls by unpaired, two-tailed Student's t-test. **f** Kaplan–Meier curves for overall survival (OS) of high ($n = 162$) versus low ADAR1 ($n = 159$) expressing cohorts in the CoMMpass (IA8) study ($n = 643$). **$p < 0.001$ by Cox regression. **g** Kaplan–Meier curves for progression-free survival (PFS) of high (top 25% ADAR1) versus low ADAR1 (bottom 25% ADAR1) expressing cohorts in the CoMMpass (IA8) study ($n = 643$), further stratified into 1q or no 1q cohorts. **$p < 0.001$ by Cox regression. Statistical significance was indicated when $p < 0.05$. See also Supplementary Fig. 1

**Table 1 Clinical information on primary samples used in this study**

| Sample | Age | Gender | Tissue | Diagnosis | Therapy | Light chain | FISH |
|---|---|---|---|---|---|---|---|
| MM1 | 64 | M | BM | Smoldering MM | Untreated | Lambda | Gain ch.5 |
| MM2 | 49 | F | BM | Smoldering MM | Untreated | Lambda | Monosomy ch.13; gain ch.5; t(14;16) |
| MM3 | 77 | M | BM | MM newly diagnosed | Supportive care | Kappa | Monosomy ch.13; del17p, t(11;14) |
| MM4 | 65 | M | BM | MM newly diagnosed | Untreated | Lambda | Gain ch.5 del17p |
| MM5 | 71 | M | BM | MM | Carfilzomib + DEX | Lambda | Loss 4p16, 17p, gain 5q, 11q, 14q |
| MM6 | 47 | M | BM | MM | CyVD | Kappa | Loss at 14q |
| MM7 | 54 | F | BM | MM relapsed | DEX[a] | Kappa | trisomy ch7 and 9 |
| MM8 | 71 | M | BM | MM relapsed | RVD[b] | Kappa | Normal |
| MM9 | 70 | M | PB | PCL | Untreated-early death | Kappa | NA |
| MM10 | 73 | M | PB | PCL | Supportive care | Lambda | NA |
| MM11 | 58 | F | PB | PCL | Unknown-lost to follow up | Kappa | NA |
| MM12 | 81 | F | BM | Smoldering MM | Untreated | Kappa | Normal |
| MM13 | 70 | M | BM | Smoldering MM | Untreated | Kappa | Monosomy ch.13, gain ch.5, 7q, 11q. |
| MM14 | 83 | M | BM | MM newly diagnosed | Velcade | Lambda | CCND1/IGH;; t(11;14) |
| MM15 | 76 | M | BM | MM newly diagnosed | Untreated | Kappa | Normal |
| MM16 | 58 | F | BM | MM relapsed | Carfilzomib + DEX previously revlimid | Kappa | CCND1/IGH; monosomy ch.13 |
| MM17 | 58 | M | BM | MM relapsed | Daratumumab/Pomalidomide/DEX | Kappa | Gain ch.1q, 5,7,9,11 |
| MM18 | 51 | F | BM | MM relapsed | Revlimid | Lambda | Normal |
| MM19 | 56 | F | BM | PCL | N/A | Kappa | Monosomy ch. 13, X; gain of 1q and 7q 13 |

Diagnosis was confirmed at the time of the bone marrow biopsy. Treatment abbreviations include: DEX, dexamethasone; CyVD, cyclophosphamide–Velcade–dexamethasone; RVD, Revlimid–Velcade–dexamethasone; NA, not available
[a]Started dexamethasone, previously on lenalidomide (revlimid), bortezomib (Velcade) and thalidomide
[b]2 years off therapy at the time of biopsy

chromosome region. ADAR1 edits adenosine-to-inosine (A-to-I) nucleotides, primarily within double-stranded (ds) RNA loops formed by primate-specific *Alu* repeat sequences[3]. Although ADAR1 p150 is expressed in response to inflammatory cytokine signaling, ADAR1 p110 is constitutively expressed[4]. Pro-inflammatory cytokine signals derived from the bone marrow (BM) microenvironment have a key role in MM progression and have been correlated with symptom severity and clonal evolution[5]. Moreover, induction of vital microenvironment-responsive stem cell self-renewal pathways, such as NOTCH1[6], promotes progression of MM. Interestingly, widespread deregulation of inflammation-responsive epitranscriptomic events also contribute to cancer stem cell (CSC) generation and maintenance, which governs cancer progression and drug resistance[7,8]. In addition to recurrent DNA mutations and epigenetic deregulation[9], RNA editing mediated by cytokine-responsive ADAR1 has emerged as a vital contributor to transcriptome remodeling leading to cancer relapse and progression[7,10–12], however the contribution of pro-inflammatory signaling leading to ADAR1-dependent RNA editing in MM pathogenesis has not been previously explored.

To address this, here we performed gene expression analysis in a cohort of primary samples with 1q amplification from the Multiple Myeloma Research Foundation (MMRF) CoMMpass study, coupled with sensitive RNA editing analyses in high-risk primary patient samples that establish robust in vivo disease models and in the setting of acquired drug resistance in vitro. Analyses revealed widespread RNA editing activation in high-risk MM patients with poor overall and progression-free survival, and robust induction of ADAR1 expression and activity in the setting of acquired immunomodulatory drug (IMiD) resistance. In contrast to ADAR1 shRNA knockdown and overexpression of an editase defective ADAR1 mutant, wild-type ADAR1 expression enhances Alu-dependent editing and transcriptional activity of GLI1, a Hedgehog (Hh) pathway transcriptional activator and self-renewal agonist, and promotes lenalidomide resistance in vitro. Finally, lentiviral shRNA ADAR1 knockdown reduces regeneration of high-risk MM in humanized serial transplantation mouse models. Thus, aberrant RNA editing represents a compelling prognostic and therapeutic target that may be exploited to obviate drug resistance in patients with MM and other therapeutically recalcitrant human malignancies.

## Results

**High ADAR1 expression in multiple myeloma predicts outcomes.** Approximately one-third of MM patients harbor amplification of chromosome 1q21[1,2], which contains ADAR1 and IL-6R loci (Fig. 1a) and is associated with a poor prognosis[13]. Although ADAR1 activation promotes progression in a broad array of malignancies[3,10,14], its role in MM pathogenesis has not been explored. To investigate if ADAR1 has a role in MM progression and relapse, we analyzed the MMRF CoMMpass database[15,16]. In 1q-amplified patients, ADAR1 levels were higher than in non-1q-amplified MM patients with the highest levels in 1q-amplified, stage III patients (Fig. 1b). In contrast to genes such as IL-6ST and ADAR2 that reside on other chromosomes, IL-6R transcripts were elevated in 1q-amplified stage I MM patients but showed no change with advancing stage (Fig. 1c; Supplementary Fig. 1a, b). Isoform-specific analyses revealed that both ADAR1 p150 and p110 were upregulated in 1q-amplified patient samples (Fig. 1d; Supplementary Fig. 1c) and in advanced-stage MM (Table 1; Fig. 1e; Supplementary Fig. 1d). Of particular prognostic relevance, a subset of samples expressing the highest levels of ADAR1 ($n = 159$) had significantly lower overall and progression-free survival rates than the low ADAR1 cohort ($n = 162$) (Fig. 1f, g; Supplementary Fig. 1e, f). Thus, in MM patients, 1q21 amplification promotes ADAR1 overexpression, and during cancer progression, as shown in other malignancies, inflammatory cytokine signaling may promote ADAR1 expression leading to poorer clinical outcomes[7,11].

**ADAR1-dependent RNA editing of GLI1 typifies high-risk myeloma.** Deregulation of ADAR1 has been linked to malignant reprogramming of progenitors into self-renewing cancer stem cells that promote progression and relapse[11,12]. Thus, we employed RNA editing site-specific quantitative real time PCR (RESSq-PCR)[17] to detect A-to-I editing, which is subsequently

read as guanosine (A-to-G), in self-renewal transcripts including the Hh pathway transcription factor GLI1. Deregulation of Hh self-renewal pathway signaling, through GLI1 and GLI2 activation, has been linked to cancer stem cell generation and

therapeutic resistance in MM[18–20] and other hematopoietic malignancies[21–23]. Previously, GLI1 transcript recoding, leading to an R701G amino acid change, was reported to stabilize GLI1 transcriptional activity by preventing the binding of a critical Hh

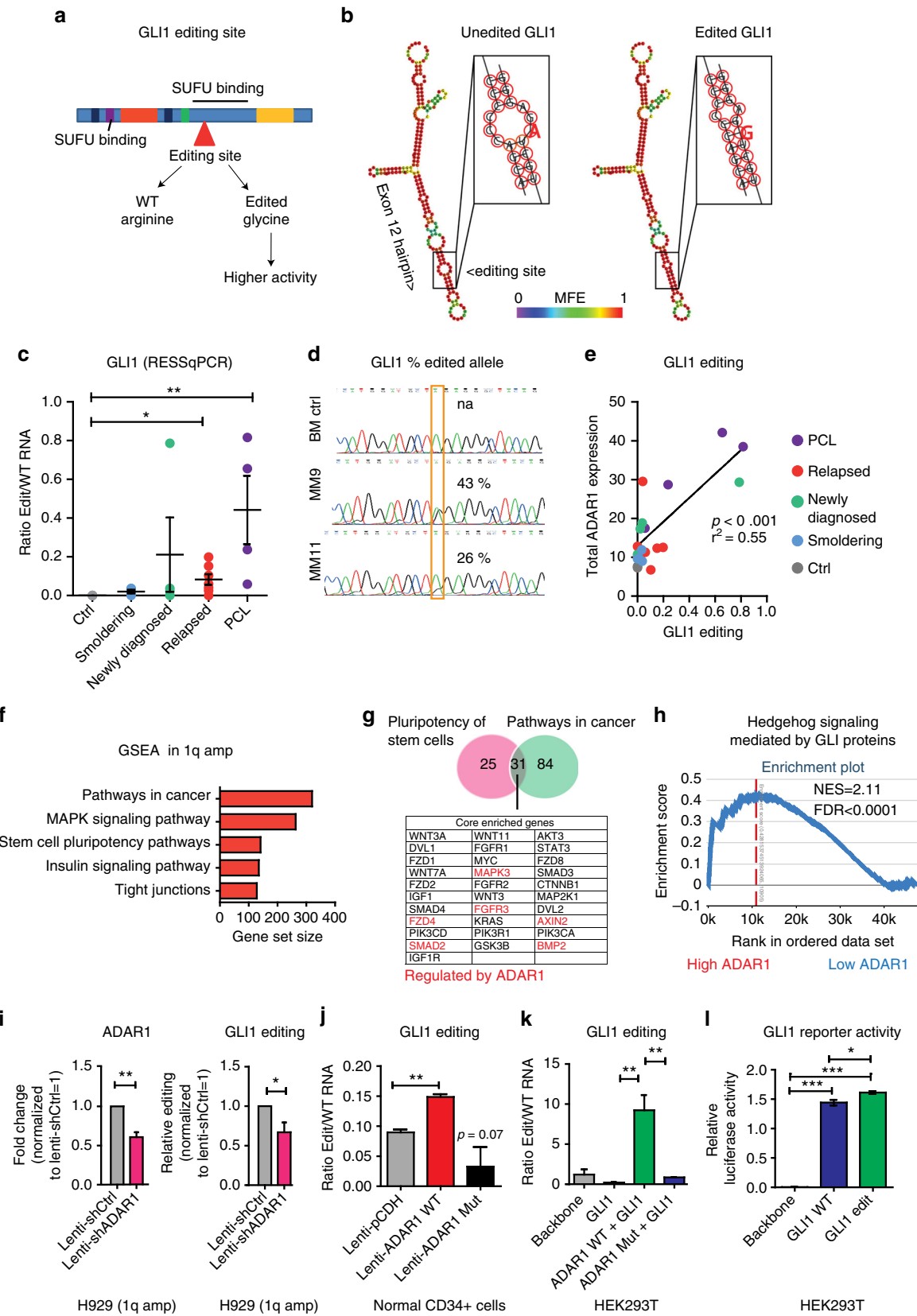

pathway negative regulator, SUFU[24] (Fig. 2a). In a Vienna software-based prediction analysis of GLI1 pre-mRNA structure, the ADAR1 edited site in exon 12 was detected in a dsRNA hairpin adjacent to a larger dsRNA structure formed by inverted *AluY* and *AluSx*, thereby altering the exonic hairpin (Fig. 2b; Supplementary Fig. 2a) and underscoring the importance of primate-specific secondary RNA structure for ADAR-mediated RNA editing[3,25]. Notably, GLI1 transcript editing rates were significantly higher in relapsed MM and PCL than age-matched controls (Fig. 2c, d), and correlated with ADAR1 expression levels (Fig. 2e). Analysis of a larger RNA-sequencing CoMMpass data set showed increased A-to-G editing of GLI1 transcripts in 1q-amplified compared with non-1q-amplified MM samples (Supplementary Fig. 2b). Gene set enrichment analyses (GSEA) of CoMMpass data revealed that 1q-amplified samples were significantly enriched in cancer and stem cell pluripotency[22] pathways, along with cell–cell adhesion[26] and drug metabolism[27] genes (Fig. 2f; Supplementary Fig. 2c). Among the KEGG-annotated pathways regulating stem cell pluripotency and pathways in cancer, several stem cell regulatory transcripts modulated by ADAR1[11], including *AXIN2*, *MAPK3*, and *FGFR3* (Fig. 2g), were enriched. Moreover, there was significant Hh pathway enrichment (NES = 2.11, false discovery rate (FDR) < 0.0001) in high ADAR1 ($n = 177$) compared to low ADAR1 ($n = 152$) patients, consistent with ADAR1-associated Hh pathway modulation (Fig. 2h). In addition, IL-6 signaling and JAK/STAT signaling were enriched in the high ADAR1 group, suggesting that inflammatory cytokine signaling may further enhance ADAR1 expression in high-risk MM (Supplementary Fig. 2d, e).

To determine whether knockdown of ADAR1 could effectively reduce GLI1 editing in ADAR1-enriched cells, we lentivirally transduced human H929 myeloma cells, which harbor 4 copies of 1q21, with shRNA targeting ADAR1 (lenti-shADAR1). Compared to control shRNA-transduced cells, lenti-shADAR1 transduced cells displayed a 40% reduction in total ADAR1 expression and a corresponding 35% reduction in GLI1 transcript editing rates (Fig. 2i). To confirm the causal relationship between ADAR1 activity and GLI1 editing, we performed lentiviral transduction of CD34$^+$ cells with ADAR1-wild-type p150 (WT), an editase defective ADAR (mutant ADAR1$^{E912A}$), or a control vector, followed by RESSq-PCR analysis. In contrast to ADAR1 mutant (ADAR1$^{E912A}$) or vector control, GLI1 transcript editing significantly increased following lentiviral ADAR1-WT transduction (Fig. 2j). We then evaluated ADAR1-dependent editing of GLI1 in co-transfection experiments in HEK293T cells, which express low levels of endogenous ADAR1 (non-1q-amplified) and GLI1. Co-transfection of ADAR1-WT and GLI1 into HEK293T cells induced robust GLI1 transcript editing (Fig. 2k). Conversely, transfection of

HEK293T cells with GLI1 and ADAR1 mutant (ADAR1$^{E912A}$) did not induce GLI1 editing, consistent with the need for intact ADAR1 RNA editase activity (Fig. 2k; Supplementary Fig. 2f). Furthermore, edited GLI1 (GLI1$^{R701G}$) induced the GLI1-*Luc* promoter reporter activity significantly more than the GLI1 WT construct (Fig. 2l), consistent with Hh signaling pathway activation observed in high ADAR1-expressing MM samples.

To further explore ADAR1-driven RNA editing events, we performed RESSq-PCR analysis on other cancer-associated loci, including the DNA cytidine deaminase APOBEC3D, antizyme inhibitor 1 (AZIN1), and murine double minute 2 E3 ubiquitin protein ligase (MDM2, Supplementary Fig. 2g–j). Consistent with previous findings in leukemia progression[11], APOBEC3D editing was significantly increased in late-stage MM patients, whereas AZIN1 and MDM2 loci showed heterogeneous editing levels. These data highlight the cell-type and context specific effects of ADAR1 editing.

**ADAR1 silencing reduces engraftment of myeloma in vivo.** Considering that increased ADAR1 expression was shown to enhance cancer stem cell generation in leukemia[11,12], we analyzed the serial transplantation potential of high-risk MM in RAG2$^{-/-}$γc$^{-/-}$ mice, as a gold standard in vivo cancer stem cell self-renewal assay[28,29], as well as the plasmacytoma-forming capacity of ADAR1-enriched MM samples (Fig. 3a; Supplementary Fig. 3a). Bioluminescent imaging demonstrated that lentiviral luciferase-expressing MM cells homed to the BM and spleen (SP) (Fig. 3b). Consistent with maintenance of malignant plasma-cell clones in vivo, patient-specific light chains were detected in the serum of the transplanted mice, within 7 to 21 weeks after transplantation (Fig. 3c). Robust engraftment of MM patient-derived cells was observed by FACS analysis in all hematopoietic tissues (Supplementary Fig. 3b). Specifically, CD138$^+$/CD319$^+$ cells[30] were detected in the BM, SP, peripheral blood (PB), and liver (L, initial site of transplantation) of RAG2$^{-/-}$γc$^{-/-}$ mice. Notably, MM9-engrafted mice developed plasmacytomas (PC) (Supplementary Fig. 3b). Consistent with previous reports[31], FACS analysis revealed that MM cells were CD38$^{high}$ and CD45$^{dim}$ (Fig. 3d; Supplementary Fig. 3c, d). In primagraft models, Alu repeat qPCR[32] detection of human-specific RNA confirmed the high rate of primary engraftment of human malignant plasma cells (Supplementary Fig. 3e, f). Moreover, high ADAR1 expression and Alu-dependent GLI1 RNA editing were also maintained in vivo (Supplementary Fig. 3e–h). Furthermore, this primagraft model propagated MM in serially transplanted recipient mice that recapitulated the engraftment phenotype observed in primary MM9-transplanted mice, as assessed by clonal light chain production in serum, and human

**Fig. 2** Aberrant RNA editing re-codes GLI1 transcripts. **a** Schematic representation of the GLI1 editing site in a putative SUFU binding domain. **b** Vienna RNA predicted secondary structure changes of GLI1 induced by A-to-I editing in exon 12. **c** RESSq-PCR analysis of GLI1 editing in primary MM total MNCs. Dots represent ratio of edit (G)/WT (A) GLI1 transcripts (mean values for individual patients ± S.E.M.; ctrl $n = 3$, smoldering MM $n = 4$, newly diagnosed MM $n = 4$, relapsed MM $n = 7$, PCL $n = 4$). *$p < 0.05$, **$p < 0.01$, by unpaired, two-tailed Student's $t$-test. **d** Representative Sanger sequencing chromatograms for GLI1; the yellow box highlights the double peak A/G, labeled with the percentage of edited transcripts assessed as edit allele burden (%G/(G + A)). **e** Regression analysis of total ADAR1 mRNA expression and GLI1 editing levels in primary samples ($n = 19$). **f** Top 5 gene sets enriched in 1q amp ($n = 40$) from CoMMpass RNA-seq (IA7 data release) ranked by gene set size. **g** Venn diagram showing number of core enriched genes between KEGG_Pathways in Cancer and KEGG_Signaling Pathways Regulating Pluripotency of Stem Cells. **h** Enrichment of Hedgehog signaling in high versus low ADAR1 patients. **i** GLI1 editing after ADAR1 silencing in 1q-amplified cells (H929). Left: ADAR1 knockdown levels by qPCR (mean ± S.E.M. of three independent experiments). Right: GLI1 editing by RESSq-PCR. *$p < 0.05$, **$p < 0.01$ compared to Lenti-shCtrl by unpaired, two-tailed Student's $t$-test. **j** GLI1 editing by RESSq-PCR in normal primary CD34$^+$ cells ($n = 3$) transduced with lentiviral pCDH ADAR1-WT/ADAR1 Mutant, or pCDH backbone control. Histograms represent relative ratios of edit (G)/WT (A) GLI1 transcripts (mean ± S.E.M. of three independent experiments). **$p < 0.01$, by unpaired, two-tailed Student's $t$-test. **k** GLI1 editing by RESSq-PCR after transient overexpression of GLI1 and WT or editase-deficient ADAR1 in HEK293T. Histograms represent mean ± S.E.M. of three independent experiments. **$p < 0.01$, by unpaired, two-tailed Student's $t$-test. **l** Relative GLI1-*Luciferase/Renilla* reporter activity (mean ± S.E.M.). HEK293T cells were co-transfected with a dual reporter plasmid and GLI1 WT, GLI1 Edit or vector control (backbone) pCDH plasmids ($n = 4$). *$p < 0.05$, ***$p < 0.001$, by one way ANOVA plus Bonferroni post-test. Statistical significance was indicated when $p < 0.05$. See also Supplementary Fig. 2

CD138[+]/CD319[+]/CD38[high]/CD45[dim] cell engraftment in BM, SP, L, and PC (Fig. 3e, f; Supplementary Fig. 3i). In serially transplanted mice, the tissues with the highest GLI1 editing also had the highest levels of human cell (*Alu*) engraftment and human

ADAR1 expression (Fig. 3g; Supplementary Fig. 3i, j). Perhaps most notably, ADAR1 silencing by lenti-shADAR1 transduction significantly reduced serial transplantation (Fig. 3h–j; Supplementary Fig. 3k, l), compared with lenti-shControl-transduced

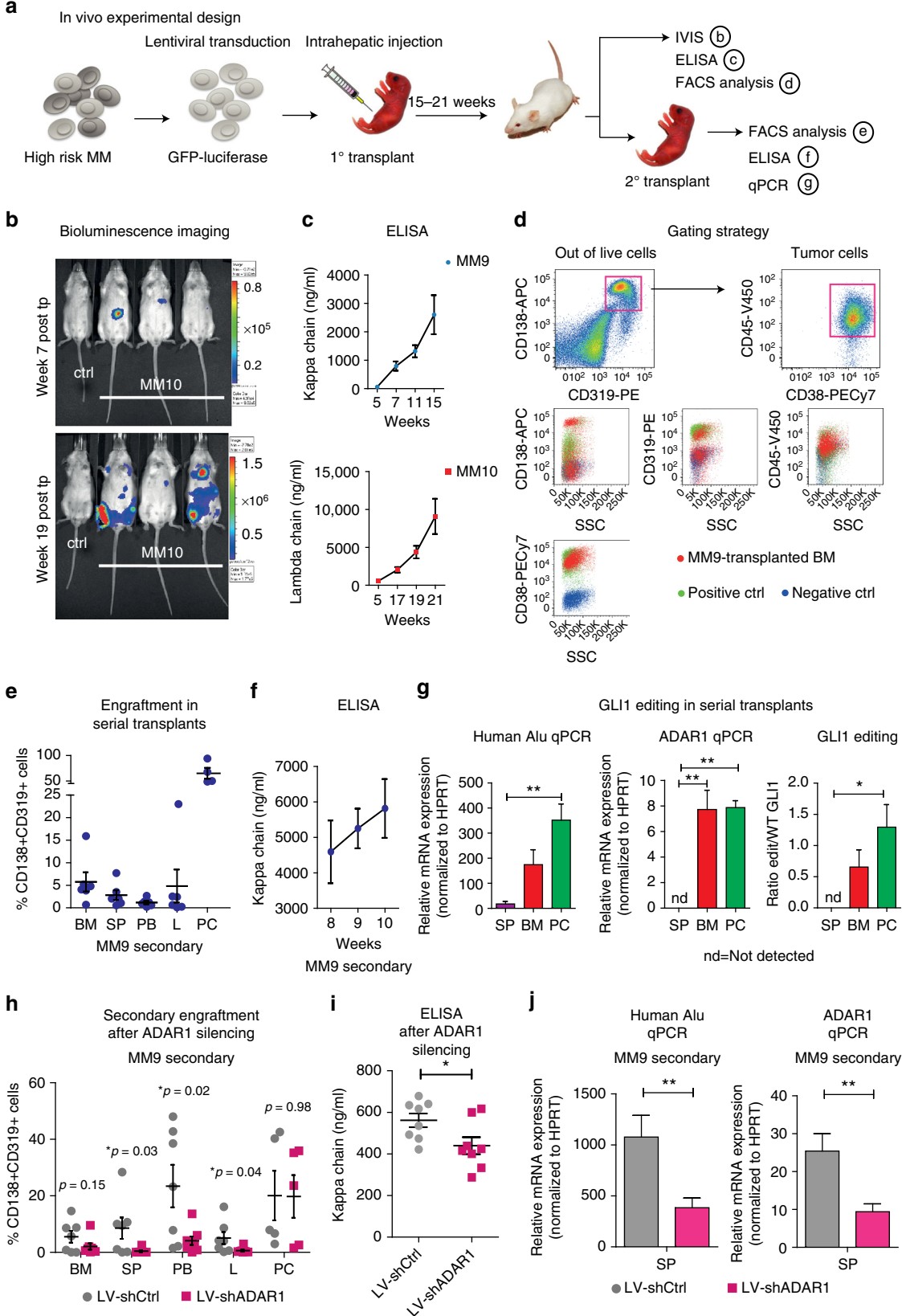

MM cells. These data are consistent with ADAR1's role in malignant self-renewal[11,12], and support a vital functional role for ADAR1 activity in malignant regeneration of MM.

**Activation of ADAR1 in myeloma cell lines promotes drug resistance.** To determine if inflammatory cytokine signaling contributes to ADAR1 activation in MM, as reported in other contexts[12,33], we treated the 1q-amplified human myeloma cell line H929 with increasing concentrations of IL-6 (0–10 ng/ml) in vitro. We observed increased ADAR1 protein accumulation and a significant increase in GLI1 RNA editing rates (Fig. 4a, b; Supplementary Fig. 4a–d). These results highlight the importance of inflammatory signaling pathways in amplification of malignant RNA editing.

Resistance to IMiDs, such as lenalidomide which disrupts cytokine-responsive pathways, is associated with MM progression. Hypothesizing that lenalidomide could modulate ADAR1 expression and activity, we generated lenalidomide-resistant H929 myeloma cell lines[34]. Continuous lenalidomide exposure significantly increased ADAR1 expression (Fig. 4c), whereas IRF4 levels were reduced (Supplementary Fig. 4e). In contrast to ADAR1, ADAR2 expression was downregulated (Supplementary Fig. 4f). Moreover, the inflammatory cytokine-responsive ADAR1 p150 isoform was significantly upregulated in lenalidomide-resistant cells (Fig. 4d). In addition, lenalidomide-resistant H929 cells showed increased ADAR1 protein levels (Fig. 4e; Supplementary Fig. 4m). Although short-term (24–48 h) IMiD exposure resulted in modest, dose-dependent ADAR1 transcriptional activation, ADAR1 protein levels increased at 48 h (Supplementary Fig. 4n, o). In keeping with increased ADAR1 activation, lenalidomide-resistant cells harbored significantly increased GLI1 editing (Fig. 4f). Together, either IL-6 exposure or continuous lenalidomide treatment enhanced A-to-I editing of GLI1 as well as other cancer-associated associated transcripts, including APOBEC3D, AZIN1, and MDM2 (Supplementary Fig. 4d, g). In keeping with ADAR1's role in self-renewal[11], lenalidomide-resistant H929 cells displayed increased replating capacity (Fig. 4g) commensurate with GLI1 transcript editing (Fig. 4h). Following lentiviral shRNA-mediated ADAR1 silencing (Supplementary Fig. 4h, i), GLI1 transcript editing (Fig. 4i) and self-renewal (Fig. 4j) were significantly reduced. Similar effects were observed with lentiviral shRNA knockdown of GLI1 (Supplementary Fig. 4j–l), thereby supporting a role for ADAR1-dependent RNA editing of GLI1 in IMiD resistance.

## Discussion

Seminal studies demonstrate that MM is typified by widespread genetic heterogeneity and subclonal diversity as well as Hh pathway activation that contribute to therapeutic resistance[18,35,36]. Compared with ADAR2[24,37], ADAR1 is the most abundantly expressed and active RNA editase in MM, consistent with potentiation of malignant regeneration by ADAR1. In MM, here we demonstrate that malignant regenerative capacity is associated with ADAR1-mediated recoding of the self-renewal agonist GLI1, a pathway which has therapeutic potential in cancer while sparing normal stem cell maintenance[23,38]. Although the Hh signaling pathway has been associated with cancer stem cell generation, and both the CD138[+19,20] and CD138[−18] cell populations secrete Hh ligands that activate autocrine signaling in MM, we have identified a primate-specific mechanism of Hh pathway activation in MM through enhanced transcriptional activation of GLI1 following *Alu*-dependent ADAR1-mediated RNA editing. Importantly, ADAR1-dependent RNA recoding of GLI1 and regeneration of MM cells harboring 1q21 amplification could be reversed through genetic knockdown of ADAR1. Overall, these findings demonstrate that both genetic (1q21 amplification) and microenvironmental (inflammatory cytokines, IMiDs) factors modulate A-to-I editing activity, which drives GLI1-dependent malignant regeneration in MM (Fig. 4k). Thus, ADAR1 activation represents both a vital prognostic biomarker and therapeutic vulnerability in MM.

## Methods

**CoMMpass study data analysis.** Gene expression data in FPKM were obtained from the CoMMpass RNA-Seq data sets (see "Data Availability"). In this study, ADAR1, IL6R, IL6ST, and ADAR2 RNA-Seq data from CD138-purified cells was included (n = 529; 1q n = 100; no 1q n = 429; IA8 data release); patients were further stratified among the groups 1q amplification versus no 1q according to the International Staging System[39], when available (1q stage I n = 29, stage II n = 39, stage III n = 32; no 1q stage I n = 146, stage II n = 150, stage III n = 133). ADAR1 isoforms expression analyses were evaluated in a smaller subset of samples (1q n = 39; stage I n = 14; stage II n = 17, stage III n = 8; no 1q n = 47, stage I n = 20, stage II n = 14, stage III n = 13; IA7 data release).

**Primary sample processing.** MM patient samples and normal age-matched (n = 3, mean age = 60.6 ± 16.8 years old) BM samples were obtained from consenting patients in accordance with Institutional Review Board approved protocols at University of California-San Diego (UCSD or the Princess Margaret Hospital (Toronto, Ontario, Canada). Peripheral blood (PB) or BM samples were processed by Ficoll density centrifugation in a SepMate conical tube (StemCell Technologies). Viable total mononuclear cells (MNC) were collected for further analyses and stored in liquid nitrogen.

**RNA editing site-specific quantitative PCR.** RESSq-PCR assay primer design[17] was carried out for specific cancer and stem cell-associated loci. For these assays, allele-specific primers were designed using Primer1, generating two outer and two inner primers for each editing site with melting temperatures ranging from 60 to 68 °C. The forward (FW) outer and reverse (REV) outer primers flank the editing site and can be used for Sanger sequencing validation of each editing site, and also as a qPCR positive control to ensure that the editing region is detectable in cDNA. The

**Fig. 3** High ADAR1-expressing cells serially transplant MM in primary patient-derived xenografts. **a** Schematic diagram showing primary high-risk MM (plasma-cell leukemia) patient-derived mouse model generation and analyses performed. **b** Representative images of primary MM10-engrafted mice, by in vivo bioluminescence assay. Mice were injected intra-peritoneally (i.p.) with luciferin (150 mg/kg) and luciferase signal was acquired by IVIS within 10 min of injection. **c** ELISA for human immunoglobulin light chain levels was used to monitor tumor cell growth in vivo over time. **d** Representative gating strategy on transplanted mouse BM. Upper dot plots show human malignant plasma cells gated out of total single live cells. Cell surface marker expression was determined as CD138[+]/CD319[+]/CD38[high]/CD45[dim], as displayed in overlaid plots. Red dots show representative MM9-engrafted mouse BM, green and blue represent positive, and negative staining controls, respectively. **e** Human CD138/CD319 double positive cell engraftment in serially transplanted recipients (MM9, n = 6). **f** Human immunoglobulin kappa chain levels in mouse sera from M9 serially transplanted mice (n = 6). **g** qPCR analyses and GLI1 editing in serial transplants. Left panel: human cell engraftment, determined by human-specific *Alu* qPCR; center panel: ADAR1 RNA expression levels by qPCR. Right panel: GLI1 editing by RESSq-PCR. Histograms represent mean values ± S.E.M. from individual mice in spleen (SP, purple, n = 3), bone marrow (BM, red n = 4), and plasmacytomas (PC, green, n = 4) from MM9 serially transplanted animals. *p < 0.05, **p < 0.01 by one way ANOVA. **h** Engraftment of CD138/CD319 double positive cells in secondary recipients of Lentiviral (LV)-shADAR1 versus shCtrl-transduced MM9 cells (n = 7 LV-shCtrl; n = 8 LV-shADAR1). **i** ELISA for human immunoglobulin light chain levels in secondary recipients of LV-shADAR1 versus shCtrl-transduced MM9 cells (n = 7 LV-shCtrl, n = 8 LV-shADAR1; 7 weeks post intrahepatic transplant). p values by unpaired, two-tailed Student's t-test. **j** ADAR1 and human *Alu* mRNA levels by qPCR in spleen tissue of MM9 secondary LV-shCtrl or shADAR1 mice (LV-shCtrl n = 7; LV-shADAR1 n = 8). **p < 0.01 by unpaired, two-tailed Student's t-test. Statistical significance was indicated when p < 0.05. See also Supplementary Fig. 3

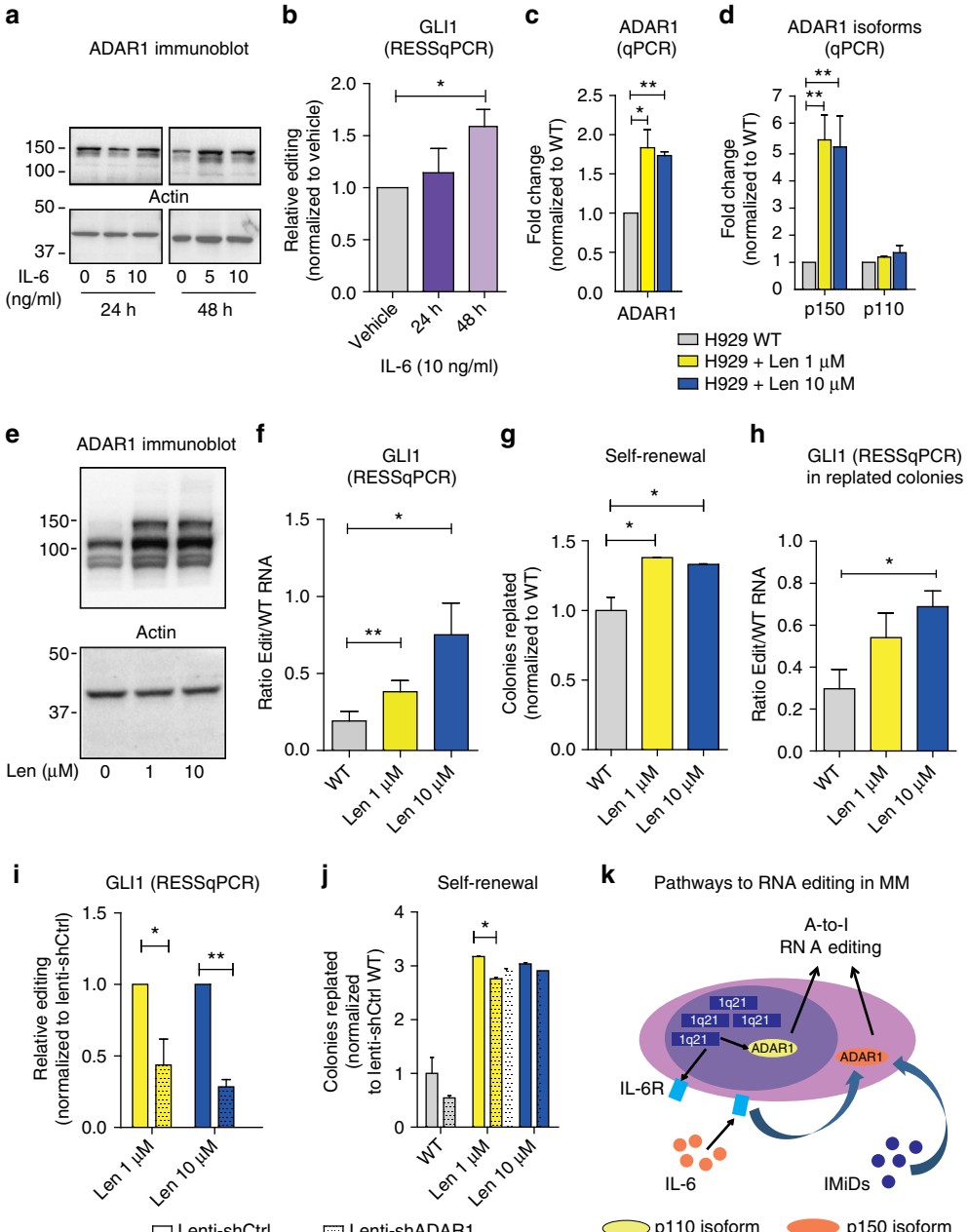

**Fig. 4** MM-niche-derived cues trigger ADAR1 expression and activation. **a** Representative Western blot analysis of wild-type (WT) H929 cells after IL-6 stimulation. Upper blot: anti-ADAR1 monoclonal antibody reveals bands at 150 kDa, corresponding to isoform p150, at 24 h (left panel) or 48 h (right panel), after IL-6 stimulation in H929, compared to vehicle control. Bottom blot: actin for normalization. **b** GLI1 RNA editing by RESSq-PCR in IL-6-stimulated cells after 24–48 h, normalized to vehicle control. Histograms represent mean ± S.E.M. of at least three independent experiments. **c** ADAR1 increased mRNA expression in lenalidomide-treated (1 μM, 5–50 weeks; 10 μM, 5–40 weeks) H929 cells compared to WT. **d** ADAR1 isoform expression in lenalidomide-treated H929 versus WT. Histograms represent mean ± S.E.M. values of at least three independent experiments. *$p < 0.05$, **$p < 0.01$ by unpaired, two-tailed Student's $t$-test. **e** Representative Western blot analysis of H929 cells under prolonged lenalidomide treatment. Upper blot: anti-ADAR1 polyclonal antibody reveals two bands at 150 kDa and 110 kDa corresponding to isoforms p150 and p110 in lenalidomide-treated cells (1 μM Len, 20 weeks; 10 μM Len, 10 weeks). Bottom blot: actin for normalization. **f** GLI1 mRNA editing in lenalidomide-treated H929 (1 μM; 10 μM) compared to WT by RESSq-PCR in at least three independent qPCR experiments. *$p < 0.05$, **$p < 0.01$ by unpaired, two-tailed Student's $t$-test. **g** Colony formation assays of Len-resistant H929 versus WT (mean ± S.E.M., $n = 3$ per condition). Histograms show relative colony replating capacity compared to WT; *$p < 0.05$ by unpaired two-tailed Student's $t$-test. **h** GLI1 RNA editing by RESSq-PCR in replated colonies from Len-resistant H929 versus WT (mean ± S.E.M., $n = 3$ per condition); *$p < 0.05$ by unpaired two-tailed Student's $t$-test. **i** GLI1 RNA editing by RESSq-PCR after silencing (Lenti-shCtrl versus Lenti-shADAR1) in lenalidomide-resistant H929 cells. Histograms show mean ± S.E.M. *$p < 0.05$, **$p < 0.01$ by unpaired, two-tailed Student's $t$-test. **j** Self-renewal of Len-resistant versus WT H929 upon lentivirally-enforced silencing of ADAR1. Histograms show relative colony replating compared to shCtrl WT (mean ± S.E.M., $n = 3$ per condition). *$p < 0.05$ by unpaired two-tailed Student's $t$-test. **k** Schematic representation of ADAR1>-RNA editing axis promoting drug resistance and tumor cell survival. Statistical significance was indicated when $p < 0.05$. See also Supplementary Figs. 4 and 5

3′ ends of the FW inner and REV inner primers match either the WT A or edited G nucleotide, and include an additional mismatch two nucleotides upstream of the 3′ primer end to enhance allelic discrimination. Relative RNA editing rates (Relative Edit/WT RNA) were calculated using the following calculation: $2^{-(Ct\ Edit\ -\ Ct\ WT)}$.

**Gene set enrichment analyses and double strand structure prediction**. Gene expression values (FPKM) downloaded from the MMRF CoMMpass study web portal (https://research.themmrf.org and study accession phs000748.v4.p3) were input into the GSEA software[11], and through the MMRF GSEA algorithm with NCI-provided gene sets. Enriched gene sets were derived from the analysis with a FDR < 10% and $p$ value < 0.05.

GLI1 editing analysis was performed on whole transcriptome data publicly available from the CoMMpass study. A total of 97 samples were included. RNA-seq raw reads were aligned to reference genome GRCh37. Aligned reads were processed with SAMTools, and editing (A or G SNV) was evaluated in GLI1[17]. Putative A-to-G events with equal or higher than 2 reads were included in the final analysis ($n = 14/17$ with/without 1q21 amplifications, respectively). Consistent low coverage was observed among all samples for the site. Ratio of G over total reads is displayed in results. Secondary structure of GLI1 transcript were predicted by Vienna RNA software[40].

**Lentiviral transduction**. Prior to transplant (primary MNCs) or in vitro experiments, cells were plated in U-bottom 96-well plate and incubated with lentivirus for 48 h in regular media (H929) or StemPro (ThermoFisher) complete media (primary MNCs). Lentiviral GFP-luciferase (GLF) encoding vector was used at 100 Multiplicity of Infection (MOI); lentiviral shGLI1 (Genecopoeia)/shADAR1[12] (Thermofisher) or relative shCtrl backbone controls were used at 50–100 MOI. An editase-deficient ADAR1 mutant (ADAR1E912A)[11] or ADAR1-WT lentiviral vectors in pCDH-EF1-T2a-GFP backbone were used in primary cells at MOI = 100.

**Transient GLI1 and ADAR1 overexpression**. HEK293T cells were plated in 12-well plates ($0.25 \times 10^6$ cells per well) 24 h prior transfection. pCDH backbone or pCDH-GLI1 plasmids were used for transient transcript overexpression, co-transfected with pDEST ADAR1-WT or mutant[11] ADAR1E912A. 1 µg of total plasmid DNA was used for each condition and cells were collected 48 h post-transfection for further analyses.

**Dual reporter assays**. HEK293T cells were plated in 96-well plates ($0.3 \times 10^5$ cells per well) 24 h prior to transfection with Lipofectamine 2000 (Life Technologies). pCDH backbone/GLI1 WT/GLI1R701G plasmids were co-transfected with a dual GLI reporter system (6X-GLI1-Luciferase reporter and Renilla reporter, 1:1 ratio) for transient transcript overexpression. Cells were collected 48 h post-transfection for further analyses of promoter activation by luminescence detection.

**Intrahepatic inoculation of tumor cells and tissue collection**. All animal studies were performed in accordance with UCSD and NIH-equivalent ethical guidelines and were approved by the university institutional animal care and use committee (IACUC). Newborn (1–3 days) Balb/c Rag2−/−γc−/− mice (sample size depending on litter survival rates) were intrahepatically injected with a 30 gauge Hamilton syringe (Hamilton Company). Each animal received $1–2 \times 10^6$ MNCs isolated from primary samples. Animals were weaned at 3 weeks of age and monitored regularly by health status assessment; in vivo bioluminescence imaging (IVIS 200) and peripheral blood screening were regularly performed until signs of disease were observed, including significant loss of weight, limited mobility and presence of palpable tumors. Mice were killed by $CO_2$ inhalation. PB was collected by cardiac puncture immediately after sacrifice. Bones, SP, L, and PC were collected in ice cold HBSS containing 2% fetal bovine serum (FBS). For serial transplantation assays, hematopoietic tissues and PC were processed into single-cell suspensions, enriched for human cells using a mouse cell depletion kit (Miltenyi), and equal proportions of viable bone marrow and PC-derived human cells were mixed for transplantation into serial transplant recipient mice.

**Western blot and nanoproteomic analysis**. A total of $5–10 \times 10^6$ cells were harvested in RIPA buffer for total protein extraction. Overall, 10–20 µg of each sample were loaded onto 10% polyacrylamide gels for gel electrophoresis. Primary antibodies (polyclonal anti-ADAR1 ab88574 1:500, monoclonal anti-ADAR1 ab126745, 1:1000, and anti-actin ab8227, 1:1000, Abcam) were prepared in blocking buffer. Membranes were incubated overnight at 4 °C with primary antibodies, followed by secondary antibody incubation (anti-rabbit HRP or anti-mouse HRP, 1:5000, Cell Signaling Technology; or anti-chicken HRP, 1:5000, Abcam) in blocking buffer for 1 h at room temperature. Blots were developed using enhanced chemiluminescence (Femto Detection kit, Promega) on a Chemidoc digital imaging machine. Representative images out of three independent Western blots are shown in main and supplementary figures, with uncropped images of key blots provided in Supplementary Fig. 5. Nanofluidic experiments were performed with the Nanopro 1000 instrument (Cell Biosciences). ADAR1 was detected using an ADAR1-specific antibody (ab168809, 1:500 Abcam). A β2-microglubulin-specific

antibody (β2M; 1:500 Upstate) was used to normalize the amount of loaded protein.

**Data availability**. The RNA-Seq data that were analyzed in this study are publicly available through the Multiple Myeloma Genomics Initiative (https://research.themmrf.org) with the identifiers IA7, IA8, and IA9. These data were generated as part of the Multiple Myeloma Research Foundation CoMMpass (Relating Clinical Outcomes in MM to Personal Assessment of Genetic Profile) study (www.themmrf.org), and all linked genotype and phenotype data used have previously been deposited into the NCBI database of Genotypes and Phenotypes (dbGaP study accession phs000748.v4.p3, BioProject PRJNA248539, SRA study SRP047533). Data can be accessed via submission of a request to dbGaP at https://www.ncbi.nlm.nih.gov/gap. All other remaining data are available within the article and supplementary files, or available from the authors upon request.

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

## Acknowledgements

We wish to thank E.D. Ball and C. Mason at UC San Diego for technical assistance. This work was funded through the NCI (C.H.M.J., 1R21CA194679, R21CA189705, and 1R01CA205944), the NIDDK (1R01DK114468), the Multiple Myeloma Research Foundation (L.A.C.), the Blasker-Rose-Miah Fund of The San Diego Foundation (L.A.C.), the Leukemia & Lymphoma Society's Quest for CURES Research Grant Program (C.H.M.J., 0754-14), the Italian Ministry of Education, University and Research (E.L., doctoral fellowship, co-mentor R.C.), the Strauss Family Foundation, the Moores Foundation, the Koman Family Foundation, the Sanford Stem Cell Clinical Center, and a NCI Cancer Center Support Grant to the Moores Cancer Center (P30-CA 023100).

## Author contributions

L.A.C., E.L., R.C., and C.H.M.J. conceived of the study; and E.L., L.A.C., P.K.M., H.L., C.N.W., G.P., S.A., Q.J., and C.H.M.J., designed and/or performed experiments and analyzed data. N.D.S., L.A.C., E.L., A.P.G., and A.C.M. performed analysis of RNA-sequencing data sets from the CoMMpass study. C.H.M.J., C.C., and M.M. participated in primary patient sample identification and provided primary samples; and E.L, A.K.S., L.A.C., and C.H.M.J. wrote the manuscript with input from all authors.

## Additional information

**Competing interests:** The authors declare no competing financial interests.

