## [Peer Review File · Nature Communications]

REVIEWERS' COMMENTS:

Reviewer #1 (Remarks to the Author):

I have reviewed this manuscript in prior forms for Nature medicine and note the responses to my comments about statistical tests and repetitions. I am satisfied with the revisions but have one minor concern to point out There does seem to be an increase in ADAR1 with IL6 and increased editing and increased protein. I wasn't sure at first what band to look at in supplemental figure 4c as it looked like in figure 4a only the 150 kDa form is noted an in support figure 4c its the 110 kDa form. can you please clarify and better label the mol weight markers on Figure 4a.

We appreciate the reviewer's encouraging comments about our revisions and we now have now carefully addressed their remaining question.

Reviewer #1 (Remarks to the Author):

I have reviewed this manuscript in prior forms for Nature medicine and note the responses to my comments about statistical tests and repetitions. I am satisfied with the revisions but have one minor concern to point out There does seem to be an increase in ADAR1 with IL6 and increased editing and increased protein. I wasn't sure at first what band to look at in supplemental figure 4c as it looked like in figure 4a only the 150 kDa form is noted an in support figure 4c its the 110 kDa form. can you please clarify and better label the mol weight markers on Figure 4a.

Comments:

We have now revised Figure 4a to more clearly label the molecular weight markers. In addition, to comply with the journal's editorial formatting, we also now have supplied uncropped scans of this blot along with the blot shown in Fig. 4e as a new supplementary figure in the supplementary information. Please see new Supplementary Fig. 5.